# Study on the Function of *SlWRKY80* in Tomato Defense against *Meloidogyne incognita*

**DOI:** 10.3390/ijms25168892

**Published:** 2024-08-15

**Authors:** Yinxia Chen, Zhize Wang, Weidan Nie, Tingjie Zhao, Yule Dang, Chenghao Feng, Lili Liu, Chaonan Wang, Chong Du

**Affiliations:** 1College of Horticulture, Xinjiang Agricultural University, Urumqi 830052, China; cyx61222@163.com (Y.C.); werick201026@163.com (Z.W.); 13779334686@163.com (W.N.); 13369647866@163.com (T.Z.); 15029523014@163.com (Y.D.); fch20220515@163.com (C.F.); liulili4477@163.com (L.L.); wcn0107@126.com (C.W.); 2Postdoctoral Station of Horticulture, Xinjiang Agricultural University, Urumqi 830052, China

**Keywords:** *SlWRKY80*, *Meloidogyne incognita*, tomato, regulation, disease resistance

## Abstract

WRKY transcription factors (TFs) can participate in plant biological stress responses and play important roles. *SlWRKY80* was found to be differentially expressed in the *Mi-1*- and *Mi-3*-resistant tomato lines by RNA-seq and may serve as a key node for disease resistance regulation. This study used RNAi to determine whether *SlWRKY80* silencing could influence the sensitivity of ‘M82’ (*mi-1*/*mi-1*)-susceptible lines to *M. incognita*. Further overexpression of this gene revealed a significant increase in tomato disease resistance, ranging from highly susceptible to susceptible, combined with the identification of growth (plant height, stem diameter, and leaf area) and physiological (soluble sugars and proteins; root activity) indicators, clarifying the role of *SlWRKY80* as a positive regulatory factor in tomato defense against *M. incognita*. Based on this phenomenon, a preliminary exploration of its metabolic signals revealed that *SlWRKY80* stimulates different degrees of signaling, such as salicylic acid (SA), jasmonic acid (JA), and ethylene (ETH), and may synergistically regulate reactive oxygen species (ROS) accumulation and scavenging enzyme activity, hindering the formation of feeding sites and ultimately leading to the reduction of root gall growth. To our knowledge, *SlWRKY80* has an extremely high utilization value for improving tomato resistance to root-knot nematodes and breeding.

## 1. Introduction

Plant parasitic nematodes (PPNs) account for approximately 10% of the nematode population, and the global economic loss caused by PPNs can reach USD 157 billion per year, making them one of the most important disease agents and pests in agricultural production [1]. Root-knot nematodes (RKNs) rank first among the PPNs in terms of damage caused to agriculture [2]. The common *Meloidogyne* spp. include *M. incognita*, *M. arenaria*, *M. hapla*, and *M. javanica*, among which *M. incognita* is a particularly destructive dominant pathogenic nematode [3] and is widely distributed among a wide variety of hosts [4]. Tomato, a major cash crop worldwide, is the typical host of RKNs. During nematode infection, J2, which represents the infectious state [5], invades the host from the root tip, migrates to the vascular bundle through the intercellular space, searches for suitable feeding cells, absorbs water and nutrients from giant cells through their mouth needle, and eventually induces the occurrence of large root galls [6].

The main genes involved in resistance to RKNs in tomatoes are those of the *Mi* gene family, whose members originated from *S. peruvianum* [7]. The incompatibility of distant hybridization makes it difficult to construct populations and clones. As a result, other members of this family, except *Mi-1*, have not been fully explored [8]. Although *Mi-1* can mediate resistance and produce a hypersensitive response (HR), resistance is regulated by a variety of biological factors, including important WRKY TFs [9]. The N-terminus of the DNA-binding domain of WRKYs contains a conserved heptapeptide sequence, “WRKYGQK”, resulting in WRKYs participating in complex biological pathways by specifically binding to W-box elements within the promoter region of the targets [10]. Based on these findings, WRKYs can be modified by phosphorylation of MAPK cascade activation signals to participate in pathways associated with important disease resistance signals, including SA, JA, and ROS [11].

Previous studies have shown that endogenous SA is synthesized by the isochlorate (IC) and phenylalanine ammonia lyase (PAL) pathways [12]. For example, in *Arabidopsis*, WRKY28 directly binds to the IC synthase 1 (*ICS1*) gene promoter to activate expression and mediate SA synthesis [13]. A newly identified WRKY, BnaWSR1ca, binds directly to *ICS1* and promotes SA production, thereby controlling cell death [14]. In addition, WRKYs can directly or indirectly activate *nonexpressers of the pathogenesis-related gene* (*NPR*), tgaccg-binding factor (*TGA*) and pathogenesis-associated protein 1 (*PR1*), which are involved in SA-induced systemic acquired resistance (SAR) [15]. For instance, the interaction between CmWRKY15-1 and CmNPR1 in *Chrysanthemum morifolium* may activate the expression of the downstream *PR1/2/10* gene, thereby enhancing its resistance to *Puccinia horiana* [16]. WRKY53 can interact with TGA3 and rely on *NPR1* to activate the *CmYLCV* promoter and mediate SA signal production to improve disease resistance [17]. In contrast, *GhWRKY70* regulates the accumulation of SA signals by upregulating the expression of the *PR1* and *NPR1* genes and is involved as a negative regulator of the defense response against *Verticillium dahliae* in cotton [18]. WRKYs can also bind to the ZIM domain of jasmonic acid (JAZ) to regulate the accumulation of JA signals and mediate the resistance process [19]. In the defense response to *Phytophthora sojae*, GmWRKY40 can interact with the JAZ2/4/5/6/7/8 proteins and mediate the production of induced systemic resistance (ISR) by regulating the accumulation of H_2_O_2_ and the activity of the JA signaling cascade [20]. PnWRKY9 activates the defense of *Panax notoginseng* against *Fusarium solani* by binding to the promoter of the JA reactive defense gene *PnDEFL1* [21]. In contrast, OsWRKY72 in rice can directly bind the promoter of the JA biosynthesis gene *AOS1* and negatively regulate JA synthesis and resistance to *Xanthomonas oryzae* [22]. ROS often function as marker signals in the SA/JA cascade, and programmed cell death (PCD) mediated by ROS is the main factor involved in the defense of many plants against pathogens via HR [23,24].

In *S. lycopersicum*, researchers also explored the WRKYs’ involvement in defense against RKNs. WRKY72 TFs (WRKY72, WRKY73, and WRKY74) all actively regulate the level of resistance to RKNs and potato aphids through *Mi-1*-mediated effector-triggered immunity (ETI) processes. Among them, the direct homolog *AtWRKY72* utilizes SA-independent defense mechanisms [25]. Previous studies have found that overexpression of *SlWRKY45* can decrease the expression levels of marker genes (*PR-1* and *Pin2*) in JA and SA signals. Further studies have found that *SlWRKY45* can specifically bind to the promoter of the JA synthesis gene *SlAOC* and inhibit its expression. In addition, the susceptibility of tomato plants to RKNs was enhanced [26,27]. Another study also found that *SlWRKY3* and *SlWRKY35* were significantly induced at the tomato root feeding sites after RKN infection, and *SlWRKY3* responded positively to the SA signal. Overexpression and knockout of *SlWRKY3* indicated that it positively regulated resistance to *M. javanica* [28].

In summary, WRKYs, as a very important family of TFs, play important roles in the process of disease resistance and in regulating resistance levels together with SA, JA, and ROS signaling. The above studies have shown that WRKYs are very necessary to regulate the resistance of tomatoes to RKNs. Based on the RNA-seq results for different resistance materials (‘Motelle’ (*Mi-1*/*Mi-1*) and ‘LA3858’ (*Mi-3*/*Mi-3*)) from the previous period, *SlWRKY80* was found to be differentially expressed according to both sets of sequencing results (FC_(‘Motelle’)_ = 4.25, SRA: SRP502510, FC_(‘LA3858’)_ = 3.82, SRA: SRP355506) (Appendix A). Combined with the research of its highly homologous gene *SlWRKY70* [29], the results of a qPCR study showed that the expression level of *SlWRKY70* in the root was significantly upregulated after RKN infection, and exogenous SA could induce the accumulation of this gene. RNAi further indicated that *SlWRKY70* positively regulates the resistance level in the process of *Mi-1*-mediated resistance. Therefore, *SlWRKY80* may be involved in disease resistance to RKNs as a key regulatory node. In this study, *SlWRKY80* was selected as the research target. Through functional identification and statistical analysis of the growth and physiological indices and accumulation levels of various resistance signals, the possible disease resistance regulatory pathway of *SlWRKY80* was preliminarily identified, laying a solid foundation for further exploration of the molecular mechanism of this valuable gene.

## 2. Results

### 2.1. Silencing of SlWRKY80 Increased the Sensitivity of Tomato Plants to RKNs

The virus-induced gene silencing (VIGS) strategy was used to preliminarily verify the regulatory function of *SlWRKY80*. In a previous study, *SlWRKY80* was silenced in resistant tomato plants ‘Motelle’, and *SlWRKY80* was shown to act as a positive regulator to significantly reduce *Mi-1*-mediated resistance [30]. In this study, silencing *SlWRKY80* in susceptible tomato plants ‘M82’ again verified the directionality of its regulatory feedback. Therefore, ‘M82’-silenced individuals with a greater than 50% decrease in *SlWRKY80* expression were obtained and inoculated with *M. incognita* to detect disease resistance (Figure 1A). The results showed that, compared with those in plants not loaded with *pTRV2* and CK (‘M82’), the number of galls on the roots of ‘M82’ individuals after silencing increased, although the disease index (DI) and resistance level (RL) did not change significantly. However, overall disease susceptibility was enhanced by *SlWRKY80* silencing (Figure 1B).

### 2.2. Identification of Disease Resistance in the Overexpression Lines

*SlWRKY80* was genetically transformed into the susceptible tomato variety ‘M82’ as a receptor material, and 17 lines were obtained; of these, eight lines were successfully transformed by PCR (Figure 2A). The positive plants were identified by RT-qPCR, and the plants with relatively high *SlWRKY80* expression were selected (for which the differential expression ratio was more than 4, OE-2, OE-9, OE-15, and OE-17) for T_1_ generation reproduction for the resistance identification test (Figure 2B,C).

The test plants were inoculated with *M. incognita* at 1000 J2s per plant, after which the roots were collected and the DI was calculated (Figure 2D). The results showed that the number of root galls in overexpression plants was significantly lower than those in nontransgenic plants, and a small number of small galls with short diameters between 1 and 2 mm were distributed in the root systems of those. The diseased area accounted for 20% to 35% of the total root system. The most nontransgenic plant roots showed dense and large galls with a short diameter of approximately 5 mm. The affected area accounted for 40% to 50% of the total root system. The root-knot level of most of the overexpression plants reached 1–3, and the RL was S (DI = 62.5). The root-knot level of most of the nontransgenic plants reached 3–4, and the RL was HS (DI = 87.5). In summary, overexpression of *SlWRKY80* is beneficial for reducing the sensitivity of tomato plants to *M. incognita*.

### 2.3. Determination of the Growth Indices of the Overexpression Lines after Inoculation

The plant height, stem diameter, and leaf area of *SlWRKY80*-overexpressing T_1_ plants and nontransgenic plants were measured after inoculation with *M. incognita*. The results showed that the plant height of overexpression lines increased slowly after inoculation, while that of nontransgenic lines increased rapidly, being significantly higher than that of overexpression tomatoes at each measurement, and 1.88 times higher than that of overexpression lines at 28 days (Figure 3A). The stem diameter of the overexpression and nontransgenic lines increased slowly, and the stem diameter of the overexpression lines was always greater than that of the nontransgenic lines. There was a significant difference between the two groups at 8 d, 16 d, and 24 d after inoculation, but the stem diameters of the two treatment groups gradually became similar overall (Figure 3B). A similar trend was observed for leaf area, as there was no significant difference at each treatment stage (Figure 3C). Obviously, after *SlWRKY80* overexpression, the overall disease resistance level of the plants was improved. In particular, the plants did not show an excessive growth trend, and the relatively healthy growth state was maintained.

### 2.4. Determination of the Physiological Indices of the Overexpression Lines after Inoculation

Soluble sugar, protein, and root activity can reflect the growth state of the plant. After overexpression and nontransgenic tomatoes were inoculated, the soluble protein content exhibited a continuous trend of first increasing and then decreasing (Figure 4A). In the first 20 days after inoculation, there was no significant difference in the content of soluble protein in the overexpression lines, while in the nontransgenic lines, there was still no significant change at all stages. The overall trend of soluble protein content in overexpression lines was always higher than that in nontransgenic lines. In addition, at 4 d and 16 d, there was a significant difference between the two treatments. At 28 d, the protein content of the overexpression plants was 3.27 Ug/g FW, 2.44 times that of the nontransgenic lines.

The soluble sugar content of the overexpression lines generally decreased first and then increased (Figure 4B), with a minimum value of 0.3 μg/g FW occurring at 8 d after inoculation, after which it reached the highest level at 24 d (1.16 μg/g FW), which was 1.43 times greater than the same stage of the nontransgenic plants. At 28 d, the concentration was reduced to 1.09 μg/g FW, but compared with the nontransgenic plants, the content showed a significant difference, being 3.21 times higher. The soluble sugar content of the nontransgenic lines showed an overall increasing trend, reaching the highest value at 20 d (0.84 μg/g FW), then decreasing, and finally dropping to 0.34 μg/g FW at 28 d.

Root activity can directly reflect the degree of infestation and destruction from RKNs. After inoculation (Figure 4C), the activity in the roots of the overexpression lines was greater than that in the roots of the nontransgenic lines at all stages and reached the highest content of 18.05 mg/g/h FW at 12 d, which was 2.55 times greater than in the nontransgenic plants. At 28 d, the root activity of the overexpression plants was 17.39 mg/g FW, which was 3.38 times greater than that of the nontransgenic plants. However, the activity of the roots of the nontransgenic plants tended to decrease and was only 5.14 mg/g FW at 28 d. In conclusion, after overexpression of *SlWRKY80*, the physiological indices maintained higher levels than those of the WT, especially regarding better root activity, which reflects the disease resistance ability of plants, while maintaining better nutrient absorption and plant growth state.

### 2.5. Measurement of Endogenous Hormone Accumulation in SlWRKY80-Overexpressing Plants after Inoculation

By identifying the accumulation levels of endogenous SA, JA, ETH, and abscisic acid (ABA) in the overexpression and nontransgenic lines at 0–48 h after inoculation, it was found that the accumulation levels of the four hormones varied slightly in the nontransgenic lines (Figure 5). After 12 h in the overexpression lines, endogenous SA, JA, and ETH biosynthesis was rapidly induced, and at 24 h, the levels of both SA and ETH were significantly greater than those in the nontransgenic lines, with increases of 50.2% and 24.6%, respectively. SA was found to accumulate, reaching 89.87 ng/g at 48 h in the overexpression plants, which was extremely significantly greater than in the nontransgenic plants. The ETH signal rapidly peaked (0.44 pmol/g) at 36 h, and at 48 h, it decreased to 0.42 pmol/g, which was still extremely significantly greater than in the nontransgenic plants. JA levels peaked at 36 h (21.87 pmol/g) in the overexpression plants and were approximately twice as high as in the nontransgenic plants. Afterward, the accumulation level decreased, and at 48 h, it decreased to 18.08 pmol/g but was still extremely significantly greater than in nontransgenic plants. ABA signaling was induced and continuously increased 24 h after the inoculation of the overexpression lines, whose expression levels were extremely significantly greater than those in the nontransgenic lines at 36 h and reached 765.96 ng/g at 48 h. The above results demonstrated that *SlWRKY80* might mediate important signals, such as SA and JA, to induce defense responses against RKN invasion.

### 2.6. Determination of ROS System Levels in SlWRKY80-Overexpressing Plants after Inoculation

ROS is a marker signal indicating the degree of stress in plants, so the accumulation trend of ROS content is particularly important for the evaluation of disease resistance. Research has shown that the overall metabolism of ROS in overexpressing plants after inoculation with *M. incognita* was induced, with a significantly greater level at 24 h than in nontransgenic plants, and the level of ROS eventually reached 1495.62 ng/g at 48 h. H_2_O_2_ is the main form of ROS that plays a role in plants. In this study, at 12 h, the H_2_O_2_ content in the overexpression lines rapidly increased, and at this point, there was a significant difference compared to the nontransgenic plants. After 36 h, the accumulation of H_2_O_2_ tended to stabilize, reaching 63.61 μmol/g at 48 h, but was still extremely significantly greater than in nontransgenic plants (Figure 6).

ROS-scavenging enzymes play an important role in regulating the levels of ROS (H_2_O_2_). The determination of SOD, CAT, and POD (Figure 7) activities revealed that POD activity in the overexpression lines was rapidly induced at 24 h after inoculation, followed by a continuous increase in activity, reaching 3.14 U/g at 48 h, which was approximately 1.8 times greater than in the nontransgenic plants. In contrast, the activities of SOD and CAT showed a downward trend after inoculation; both presented extremely significantly lower activity than that of nontransgenic lines at 36 and 48 h, and at the 48 h stage, the activity of the two enzymes decreased by 40.3% and 43.2%, respectively, compared to the nontransgenic plants.

## 3. Discussion

WRKY TFs are widely involved in plant biotic stress responses. This study was based on RNA-seq data from different disease-resistant tomato plants (*Mi-1* and *Mi-3*), and *SlWRKY80*, which was differentially expressed, was used as the research object. Furthermore, RNAi and overexpression strategies were utilized to determine the function of *SlWRKY80* in tomato plants. By measuring the degree of change in disease resistance, it was clarified that *SlWRKY80* can serve as a positive regulatory factor in tomato defense against RKNs, which is clearer in terms of growth and physiological indicators. The dynamic change of soluble sugars and proteins in the *SlWRKY80*-overexpressing lines was overall greater than in the nontransgenic lines, which proved that the overall health of the *SlWRKY80*-overexpressing plants was relatively better [31]. Root activity can directly reflect the degree of root injury. The root activity of the overexpression lines remained high and relatively stable throughout the whole inoculation period, while the root activity of the nontransgenic lines was not only low but also exhibited a downward trend. A decrease in root activity over time made it difficult for the plants to absorb water and nutrients, eventually leading to a continuous abnormal increase in the plant height of the nontransgenic lines, and severe excessive growth [32].

During the disease resistance process, defense-related hormones such as SA and JA and other signals are often activated, leading to the production of HR in plants [33]. SA participates in SAR in plants [34], while JA and ETH are involved mainly in the ISR process of combating necrotrophic pathogen infection [35]. Previous studies have shown that overexpression of *CmWRKY15-1* in *chrysanthemum* can activate CmNPR1 and induce downstream disease resistance-related genes, which enhance resistance to *P. horiana* through the SAR system [16]. Overexpression of *NbWRKY40* increased the transcription levels of an SA biosynthesis gene (*ICS1*) and SAR marker genes (*PR1b* and *PR2*) and inhibited tobacco infection with TMV [36]. OsWRKY67 can activate the expression of *PR1a* and *PR10* by binding to their promoters, increasing the accumulation of SA and thereby enhancing plant resistance to leaf blast and bacterial blight disease [37]. In *Arabidopsis*, *WRKY11* triggers ISR by activating the JA signaling pathway, thereby increasing plant resistance to *Pst* DC3000 [38]. Overexpression of *GhWRKY70* in *Arabidopsis* can increase the expression of the JA synthesis-related gene *AtAOS1*, ultimately improving plant resistance to *Verticillium dahliae* [39].

In the early stage of RKN infection, the accumulation of plant disease resistance signals plays an important regulatory role in the defense against RKNs [40]. In the present study, after 12 h of infection with *M. incognita*, SA, JA, and ETH were induced in the SlWRKY80-overexpressing lines, increasing tomato resistance to varying degrees, possibly via the SAR and ISR pathways. In particular, SA and JA presented high levels of expression at 36 h and 48 h, respectively, showing almost twice the accumulation compared to the nontransgenic lines. However, when SA was consistently upregulated, JA and ETH accumulation levels tended to increase after 36 h. There was no significant difference in the accumulation of JA between the two treatments at 0–24 h after inoculation, and a significant difference did not appear until 36 h, which is consistent with the results obtained with ABA. In addition, during the early stage of inoculation, the increase in ABA concentration was not significant, and ABA induction was relatively delayed. Previous studies have shown that ABA indirectly participates in plant immune regulation by activating the expression of genes such as *C3H* and promoting the biosynthesis of SA [41]. These findings further suggest that ABA is not the main signal stimulating disease resistance. However, the specific molecular mechanism underlying the findings of this study needs further investigation. In summary, *SlWRKY80* can mediate the defense of tomato plants against RKN invasion through various resistance processes by regulating the accumulation of signaling pathways such as SA, JA, and ETH.

Plants can respond to pathogen attacks by generating oxidative stress through the continuous accumulation of ROS, which play a core role in plant immune responses. H_2_O_2_ is the most stable ROS and often serves as an intercellular and intracellular signal to trigger downstream stress and disease resistance responses [42]. In this study, the accumulation of total ROS and H_2_O_2_ in the overexpression lines inoculated with *M. incognita* showed an upward trend (0–48 h), and both showed significant differences compared to those in the nontransgenic lines after 12 h and 24 h, respectively. Taken together, these findings indicate that ROS (H_2_O_2_) are rapidly induced in roots and that high-level ROS production hinders the survival of J2s in this environment [43]. In addition, ROS can indirectly induce PCD at the site of infection by combining the above hormone signals to ultimately cause HR at the early stages of infection [44], reducing root gall production and improving plant disease resistance. This resistance mechanism can force local healthy cells to die, preventing J2s from establishing effective feeding sites and obtaining nutrients, thus preventing them from continuing to grow and reproduce [45]. Notably, excessive accumulation of ROS can cause severe cell damage and increase the sensitivity of plants to pathogens [46]. In this study, during the 0–48 h stage after the inoculation of the overexpression lines, SOD and CAT activities exhibited a decreasing trend, while POD activity exhibited the opposite trend. Together, these three factors regulate the accumulation of ROS (H_2_O_2_), ultimately leading to a steady-state level of ROS (H_2_O_2_) during the 36–48 h period, preventing ROS (H_2_O_2_)-related damage in tomato plants. Overall, *SlWRKY80* has a positive regulatory function in improving tomato resistance to RKNs and has extremely high potential for use in subsequent disease resistance breeding.

## 4. Materials and Methods

### 4.1. Plant Germplasm and Growth Conditions

The disease-resistant material ‘Motelle’ (*Mi-1*/*Mi-1*) was used for cDNA acquisition in the RNAi experiment, and the susceptible material ‘M82’ (*mi-1*/*mi-1*) was used for the construction of overexpression lines. The above materials were placed in a composite substrate consisting of peat/perlite/vermiculite at a ratio of 3:1:1, with an ambient temperature of 25 °C and a relative humidity of approximately 60%. The plants were housed under a 16 h/8 h light/dark cycle. After the plants reached the four-leaf and one-heart stage, molecular experiments and nematode inoculation tests were conducted.

### 4.2. Nematode Assay and Identification of Disease Resistance

Diseased tomato roots that were infected with *M. incognita* and contained many egg masses were selected, and the egg masses were removed and placed in a culture dish. Sterile water was added to cover the egg masses, after which the dish was incubated at 28 °C. Then, the J2-stage nematodes that had hatched were collected. The above J2s were inoculated during the four-leaf and one-heart stage of tomato plants, with an inoculation amount of 1000 J2s per individual. After 40 days of inoculation, the nematodes in the tomato root tissue were stained and observed using the acid fuchsin staining method, after which the number of root galls was calculated. Based on the number of root galls and the range of distribution, the root-knot levels were divided into a total of 6 levels, from 0 to 5 (0: root system with no galls; 1: 1–20% roots with galls; 2: 21–40% roots with galls; 3: 41–60% roots with galls; 4: 61–80% roots with galls; 5: 81–100% roots with galls). The disease index (DI) was calculated according to the following formula, and the resistance level (RL) was evaluated. DI = [∑ (number of plants at each level) × (corresponding number of levels)/(total number of surveyed plants × highest level numerical value) × 100]. The evaluation criteria for resistance were as follows: immunity (I): DI = 0, high resistance (HR): 0 ˂ DI ≤ 20, moderate resistance (MR): 20 ˂ DI ≤ 40, resistance (R): 40 ˂ DI ≤ 60, susceptible (S): 60 ˂ DI ≤ 80, and highly susceptible (HS): 80 ˂ DI.

### 4.3. Construction of RNAi and Overexpression Vectors

The CDS of *SlWRKY80* was obtained from the SGN database (https://solgenomics.net (accessed on 12 March 2022)) (Appendix A), total RNA was extracted from ‘Motelle’ tomato leaves using a reagent kit (Vazyme, Nanjing, China), and reverse-transcribed cDNA was obtained using a reagent kit (Vazyme, Nanjing, China). Using *Bam*H I as the cleavage site, specific upstream and downstream primers with a 20 bp recombinant homologous arm were designed, and the high-fidelity enzyme Phanta Max (Vazyme, Nanjing, China) was used for PCR amplification of the CDS region.

The *pTRV2* interference vector and *pCAMBIA-1300-mCherry* overexpression vector were linearized by single-enzyme digestion using *Bam*H I, after which an Infusion cloning kit (Vazyme, Nanjing, China) was used to clone *SlWRKY80*, completing the construction of the recombinant vector (Appendix A). Subsequently, the recombinant vector was transformed into *E. coli* DH5α, which was subsequently cultured in solid LB media (containing 50 µg/mL kanamycin) for 12 h in the dark. Monoclonal colonies were selected and cultured in a shaker at 37 °C and 220 rpm for 13 h. Finally, PCR identification and sequencing were performed.

### 4.4. RNAi and Overexpression Line Construction

The recombinant *pTRV2-SlWRKY80*, unloaded *pTRV2*, and *pTRV2-PDS* vectors were subsequently transformed into *Agrobacterium* GV3101, which was subsequently injected into the leaves of ‘M82’ tomato plants at the four-leaf and one-heart stage, with 60 plants treated with each. After infection, the plants were first stored at 18–21 °C. The tomatoes were cultured in darkness with a humidity greater than 80% for 72 h and then cultured under a photoperiod of 16 h/8 h at 23~25 °C and 60% humidity. After the *pTRV2-PDS* plants exhibited bleaching (after approximately 2 wks), the expression level of *SlWRKY80* was measured, and tomato plants with a silencing efficiency greater than 50% were selected for follow-up tests (10 individuals were selected for each material).

Clean and full ‘M82’ tomato seeds were selected and placed in a tissue culture bottle containing MS medium for dark culture at 25 °C for 2–3 d. After approximately 75% of the seeds were germinated, they were cultured at 23 °C with a photoperiod of 16 h/8 h. Approximately 7–9 d after normal culture, the true leaves, cotyledon, and hypocotyl were cut into 0.5 cm × 0.5 cm leaf blocks on a sterile superclean workbench and placed on preculture medium (MS + 2.0 mg/L ZT + 0.2 mg/L IAA) for low-light culture for 24–48 h. The explants were then infected with *Agrobacterium* GV3101 containing the plasmid *pCAMBIA1300-SlWRKY80-EGFP*, placed in coculture medium (MS + 2.0 mg/L ZT + 0.2 mg/L IAA), and cocultured under low light for 24–48 h. The cocultured explants were transferred successively into screening medium (MS + 2.0 mg/L ZT + 0.2 mg/L IAA + 100 mg/L Kan + 500 mg/L carbenicillin (Cb)) and bud elongation medium (MS + 1.0 mg/L ZT + 0.05 mg/L IAA + 100 mg/L Kan + 500 mg/L Cb) for continued culture, and subculture was performed once at 15–20 d. Finally, when the advected buds grew to approximately 1 cm, healthy regenerated young buds were selected, and the base calli were transferred to the root medium for culture (1/2 ms + 0.1 mg/L IAA + 50 mg/L Kan + 300 mg/L Cb) to form a complete plant.

### 4.5. RT-qPCR Assay

When the tomato materials to be tested grew to the four-leaf and one-heart stage, the plant leaves were selected and the total RNA and cDNA were obtained by the previous kit. The cDNA was diluted 10-fold and used as a template for RT-qPCR detection using Top Green qPCR Super Mix (Vazyme, Nanjing, China). Using the *SlActin* gene as an internal reference gene [47], primers were designed to determine the expression of the target gene at the transcriptional level (Appendix A). The cycling conditions were 5 min at 95 °C, followed by 40 cycles of 5 s at 95 °C and 10 s at 60 °C. Relative expression values were calculated using the 2^−ΔΔCT^ method [48]. Three biological replications, each with three technical repetitions, were performed for all reactions.

### 4.6. Determination of Plant Hormones and the ROS System

The transgenic plants and the control plants were inoculated with *M. incognita*. Leaves at different inoculation time points were selected for the determination of SA, JA, ETH, and ABA contents, while roots were selected for the determination of H_2_O_2_ level. The HPLC strategy was used for the above determination [49]. The contents of ROS and the activities of ROS-scavenging enzymes (SOD, POD, and CAT) in the roots were determined via ELISA kits (Tiangen, G0104, G0106, G0107, Beijing, China).

### 4.7. Measurement of Growth and Physiological Indices

The transgenic plants and the control plants were inoculated with *M. incognita*, after which the growth indices, including plant height, stem diameter, and leaf area, were measured. The leaves were selected for the determination of related physiological indices, including soluble protein, soluble sugar, and root activity, which were determined by Liu’s and Ma’s method [31,50]. The measurement period was once every 4 days for a total of 28 days, with three biological replicates for each treatment.

## 5. Conclusions

In this study, we selected *SlWRKY80* with highly differential expression from the RNA-seq data of different disease-resistant materials (*Mi-1* and *Mi-3*) as the research target. RNAi and overexpression strategies validated *SlWRKY80* as a positive regulator involved in tomato defense against *M. incognita*. Soluble protein and sugar and root activity reflected the improvement of overall plant disease resistance after overexpression of *SlWRKY80*, which was also reflected in the growth indices. In terms of disease resistance signals, *SlWRKY80* is likely to mediate disease resistance responses through the accumulation of SA, JA, and ETH signals, which results in the reduction of the invasion efficiency of RKNs by relying on the continuous accumulation of ROS.

## Figures and Tables

**Figure 1 ijms-25-08892-f001:**
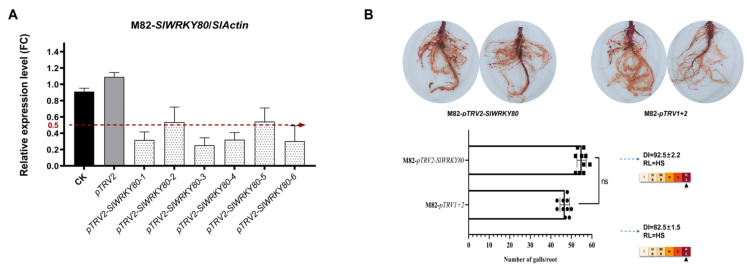
The function of *SlWRKY80* in the defense against RKNs was preliminarily verified by RNAi. (**A**) Silencing efficiency test (partial individuals), using ‘M82’ as controls (CK), was performed for RT-qPCR detection of *SlWRKY80* in both empty and silenced strains. The red dotted line represents a decrease in relative expression to 50%; (**B**) ‘M82’ were inoculated with J2s after silencing *SlWRKY80*. Statistics on gall numbers: the results are presented as the means ± SEM (n = 10, *p* < 0.05), through one-way ANOVA statistics. DI calculation and RL of the silenced and unloaded *pTRV2* strains were identified.

**Figure 2 ijms-25-08892-f002:**
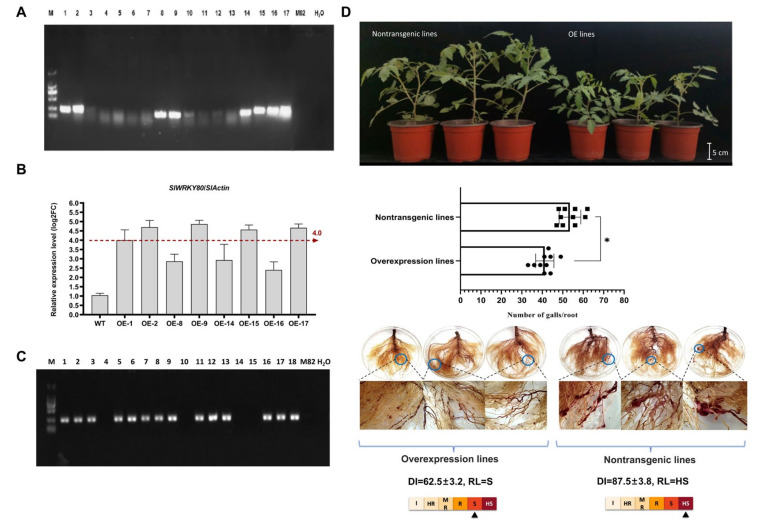
Identification of disease resistance in ‘M82’ after overexpression of *SlWRKY80*. (**A**) Positive identification of 1–17 transgenic lines. Marker: 2000 bp, the target band size is 249 bp. Among them, 1, 2, 8, 9, and 14–17 were positive lines. (**B**) The expression of *SlWRKY80* in positive plants was detected by RT-qPCR. The red dotted line indicates that the difference is more than 4 times expressed; (**C**) OE-2, OE-9, OE-15, and OE-17 transgenic plants were screened for T_1_ generation propagation. Marker: 2000 bp, the target band size is 249 bp. (**D**) After overexpressing *SlWRKY80*, individuals were inoculated with 1000 J2s. After 40 d, plant morphology and root gall morphology were observed, DI was counted, and RL was identified. Root gall statistics are presented as the means ± SEM (n = 10, * *p* < 0.05), through one-way ANOVA statistics.

**Figure 3 ijms-25-08892-f003:**
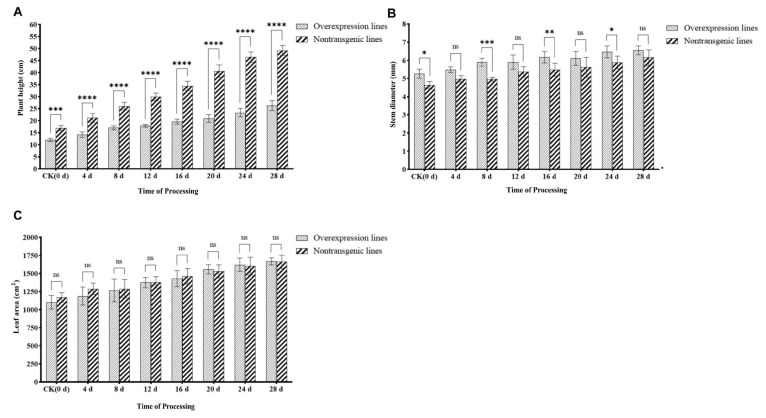
Determination of growth index of overexpression lines after inoculation. The horizontal coordinate represents the processing time of 0–28 d, and the vertical coordinate represents (**A**) plant height. The difference multiples of nontransgenic lines and overexpression lines before and after inoculation at 4 d, 8 d, 12 d, 16 d, 20 d, 24 d, and 28 d were, respectively, 1.42, 1.49, 1.53, 1.67, 1.76, 1.94, 1.99, and 1.87 times; (**B**) stem diameter; (**C**) leaf area. The results are presented as the means ± SEM (n = 3, * *p* < 0.05, ** *p* < 0.01, *** *p* < 0.001 and **** *p* < 0.0001), through one-way ANOVA statistics.

**Figure 4 ijms-25-08892-f004:**
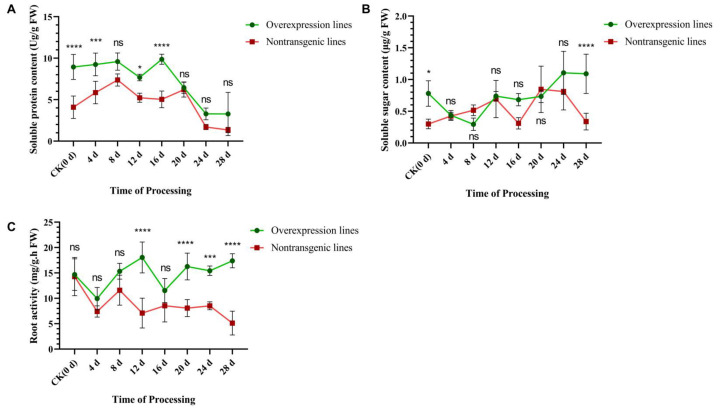
Determination of physiological index of overexpression lines after inoculation. The horizontal coordinate represents the processing time 0–28 d, and the vertical coordinate represents (**A**) soluble protein; (**B**) soluble sugar; (**C**) root activity. The lines of different colors represent significant differences in comparison with CK (0 d) at different stages after inoculation in the two treated lines, respectively. The results are presented as the means ± SEM (n = 3, * *p* < 0.05, *** *p* < 0.001 and **** *p* < 0.0001), through one-way ANOVA statistics.

**Figure 5 ijms-25-08892-f005:**
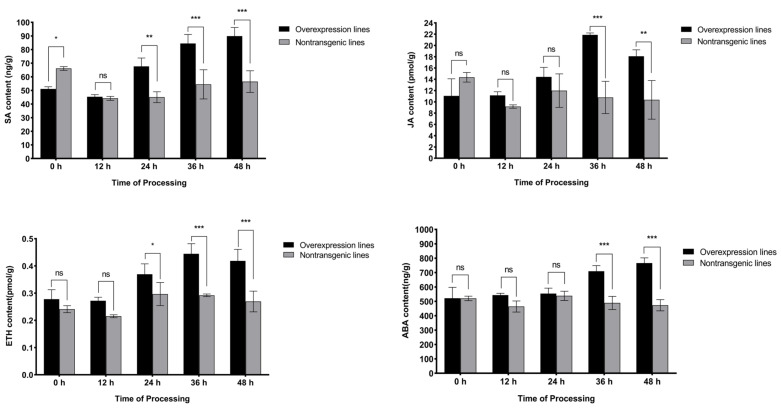
Statistics of SA, JA, ETH, and ABA accumulation in overexpression lines. The horizontal coordinate represents the processing time of 0–48 h, and the vertical coordinate represents the content of SA, JA, ETH, and ABA. The results are presented as the means ± SEM (n = 3, * *p* < 0.05, ** *p* < 0.01 and *** *p* < 0.001), through one-way ANOVA statistics.

**Figure 6 ijms-25-08892-f006:**
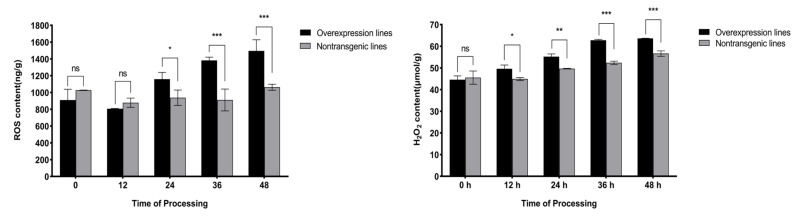
Statistics of the ROS system in overexpression lines. The horizontal coordinate represents the processing time of 0–48 h, and the vertical coordinate represents the content of total ROS and H_2_O_2_. The results are presented as the means ± SEM (n = 3, * *p* < 0.05, ** *p* < 0.01 and *** *p* < 0.001), through one-way ANOVA statistics.

**Figure 7 ijms-25-08892-f007:**
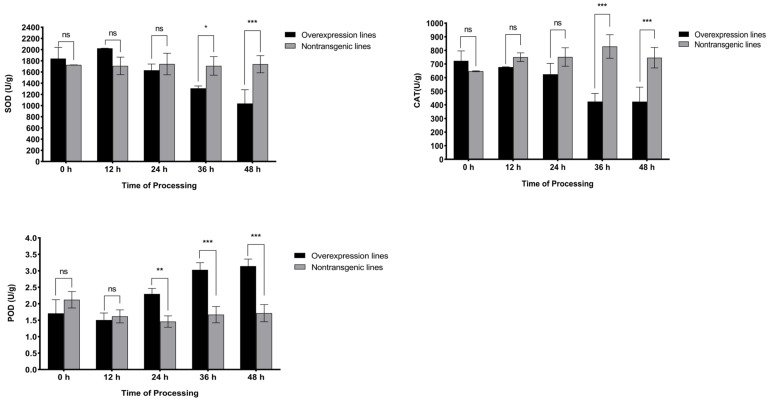
Statistics of the ROS system in overexpression lines. The horizontal coordinate represents the processing time of 0–48 h, and the vertical coordinate represents the activity of POD, SOD, and CAT. The results are presented as the means ± SEM (n = 3, * *p* < 0.05, ** *p* < 0.01 and *** *p* < 0.001), through one-way ANOVA statistics.

## Data Availability

The original contributions presented in the study are included in the article/Appendix A, further inquiries can be directed to the corresponding author.

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
