# Peer review of "Study on the Function of SlWRKY80 in Tomato Defense against Meloidogyne incognita"

_ijms, 2024, doi:10.3390/ijms25168892_

Round 1

Reviewer 1 Report

Comments and Suggestions for Authors

Chen et al. investigated the role of the SlWRKY80 gene in tomato disease resistance by both silencing and overexpressing it, then inoculating the plants with M. incognita to assess their phenotypes. They measured various growth indices and hormone levels in both wild-type (WT) and experimental groups. The study found that overexpressing SlWRKY80 appeared to enhance disease resistance in tomatoes, whereas silencing SlWRKY80 did not produce significant results. Despite measuring several growth indices and hormone levels, the authors did not elucidate the underlying connection between these indices and disease resistance. They only demonstrated a correlation between SlWRKY80 overexpression and the measured indices without uncovering specific pathways or detailed mechanisms. This lack of connection makes the manuscript appear as though it contains two independent studies: one on disease resistance and the other on growth indices.

Overall, the manuscript should better integrate the findings from disease resistance and growth indices studies, demonstrating a clearer connection and logical flow between these aspects.

Below are some specific suggestions.

Line43: what ‘HR’ stands for? Please show the whole name before you use the abbreviation.

Line56: ‘PR gene’ is pathogenesis-associated protein?

Line51-75: What do the authors try to present in this paragraph? The association between WRKY and SA pathways? Or WRKY contributes to disease defense? The authors provide too many examples here, but I could not get the key.

Line94:’ Combied’ do the authors mean ‘combined’?

Line95: Provide more detailed information on how SlWRKY70 contributes to disease resistance if using it to infer the function of SlWRKY80 due to homology. Explain the connection if GhWRKY70 is mentioned as homologous to SlWRKY70.

Line109-114: Reassess the effectiveness of silencing SlWRKY80. A 50% reduction might still allow sufficient expression for disease resistance, potentially explaining the non-significant results in gall counting between WT and silenced groups.

Line113: please provide full name of ‘DI’ and ‘RL’.

Line151-198: The authors present the dynamics of growth between WT and experimental group. What is the conclusion for these two sections? How do these results relate to the disease defense?

Line233-242: Explain the implications of higher ROS levels in the overexpression group. Does this indicate increased stress levels in these plants?

Comments on the Quality of English Language

Need to carefully polish

Author Response

Dear Editor and Reviewer1:

Thank you for your letter and for the reviewer’s comments concerning our manuscript entitled “Study on the function of SlWRKY80 in tomato defense against Meloidogyne incognita”. (ID: ijms-3111864). Those comments are all valuable and very helpful for revising and improving our paper, as well as the important guiding significance to our researches. We have studied comments carefully and have made correction which we hope meet with approval. Revised portion are marked in yellow in the paper. The main corrections in the paper and the responds to the reviewer's comments are as flowing:

Total comments: Chen et al. investigated the role of the SlWRKY80 gene in tomato disease resistance by both silencing and overexpressing it, then inoculating the plants with M. incognita to assess their phenotypes. They measured various growth indices and hormone levels in both wild-type (WT) and experimental groups. The study found that overexpressing SlWRKY80 appeared to enhance disease resistance in tomatoes, whereas silencing SlWRKY80 did not produce significant results. Despite measuring several growth indices and hormone levels, the authors did not elucidate the underlying connection between these indices and disease resistance. They only demonstrated a correlation between SlWRKY80 overexpression and the measured indices without uncovering specific pathways or detailed mechanisms. This lack of connection makes the manuscript appear as though it contains two independent studies: one on disease resistance and the other on growth indices.

Overall, the manuscript should better integrate the findings from disease resistance and growth indices studies, demonstrating a clearer connection and logical flow between these aspects.

Response: Many thanks to the reviewer’s suggestions on the manuscript. The purpose of this study is to investigate the regulatory function of SlWRKY80 in tomato after inoculation with M. incognita, including the evaluation of hormone signals, ROS signals, growth and physiological indicators. The aim is to further explore the resistance regulation mechanism of SlWRKY80 in the later stage, including the integration from the protein level (Y2H, BiFC and Pull-down technologies) and the regulatory level (Chip-seq and Dual-luc technologies). Therefore, this study is to prove the research and utilization value of SlWRKY80, and provide a basis for further exploration of its molecular mechanism.

1.Line43: what ‘HR’ stands for? Please show the whole name before you use the abbreviation.

Response: Considering the reviewer's suggestion, we added the abbreviation (line 43).

2.Line56: ‘PR gene’ is pathogenesis-associated protein?

Response: Thank you very much for your query. What we're talking about here is the ‘nonexpressers of the PR gene’ (line 56). These genes, such as NPR1, is the key factor in SA-mediated systemic acquired resistance (SAR) and play an important role in disease resistance.

3.Line51-75: What do the authors try to present in this paragraph? The association between WRKY and SA pathways? Or WRKY contributes to disease defense? The authors provide too many examples here, but I could not get the key.

Response: Thanks to the reviewer's comments. In the previous paragraph, we described the processes that WRKYs are often involved in regulating disease resistance, including SA, JA, and ROS (line 47-50). Therefore, in this part (line 51-75), we focus on SA and JA signals, which are very important signaling systems in plant disease resistance. They mediate SAR and ISR respectively, and introduce the key factors regulated by WRKYs in above signals, including NPR, PR and TGA (SA signals), JAZ and AOS (JA signals). At the same time, ROS accumulation is often associated with the above regulation process to affect the level of disease resistance. Therefore, we added related reference to support it. To sum up, the description of this part is in line with scientific logic.

4.Line94:’ Combied’ do the authors mean ‘combined’?

Response: We are very sorry for the incorrect spelling. We have changed it (line 94).

5.Line95: Provide more detailed information on how SlWRKY70 contributes to disease resistance if using it to infer the function of SlWRKY80 due to homology. Explain the connection if GhWRKY70 is mentioned as homologous to SlWRKY70.

Response: Thanks to the reviewer's valuable suggestions. According to the suggestions, we supplemented the functional exploration of SlWRKY70 to provide a reasonable and solid basis for our research on SlWRKY80 (line 95-100).

6.Line109-114: Reassess the effectiveness of silencing SlWRKY80. A 50% reduction might still allow sufficient expression for disease resistance, potentially explaining the non-significant results in gall counting between WT and silenced groups.

Response: Thanks very much for your query. First, we believe that a silence efficiency of less than 50% can confirm the effectiveness of VIGS, which has been confirmed in previous studies (Tao Wang et al., 2024. SlNAC3 suppresses cold tolerance in tomatoes by enhancing ethylene biosynthesis. Plant, Cell & Environment. 47(8), 3132-3146). The reason why the number of root galls is not significant is that the 'M82' we used is a highly sensitive material (line 342), which also corresponds to our overexpressed tomato material. Therefore, in high-sensitivity materials, the difference in the number of root galls may not be very obvious even if SlWRKY80 silenced, but there is a difference (line 116-117), we also used enough silenced individuals to verify (line 390-391), which precisely shows that SlWRKY80 may have a positive regulation in regulating resistance to RKNs.

7.Line113: please provide full name of ‘DI’ and ‘RL’.

Response: According to the reviewer's suggestion, we have supplemented the full name (line 117).

8.Line151-198: The authors present the dynamics of growth between WT and experimental group. What is the conclusion for these two sections? How do these results relate to the disease defense?

Response: Thanks very much for your query. In this study, the growth (plant height, stem diameter and leaf area) and physiological indicators (soluble sugar, soluble protein and root activity) of SlWRKY80 overexpressed strain (T1) were measured, aiming to prove that SlWRKY80 could reduce the generation of root galls while maintaining a good growth state of the whole plants, which comprehensively indicated its value in improving disease resistance (line 156-210). Therefore, in results section of the manuscript, we performed a description of the assay results. According to the reviewer's suggestion, we conducted a comprehensive discussion in the discussion part, together with resistance measurement and signal accumulation identification. For example, line 271-282, we explained that SlWRKY80 overexpressed maintained good root activity of tomato plants and ensured nutrient absorption, thus maintaining reasonable physiological and growth indicators. In the conclusion section, we also added the explanation of the results of this part (line 435-437).

9.Line233-242: Explain the implications of higher ROS levels in the overexpression group. Does this indicate increased stress levels in these plants?

Response: Thanks very much for your query. First of all, the importance of ROS signal in disease resistance was illustrated in the section of ROS determination of overexpressed strains (line 238-239). In the discussion section (line 317-338), we discuss the consistency of this study combined with previous studies. As a core signal, ROS inhibits the formation of feeding sites of J2s (RKNs), and can induce programmed cell death (PCD) to form HR. Therefore, the accumulation and balance of ROS are the key to defense against RKNs infection.

Thanks again to the reviewer for taking time out of their busy schedules to review our manuscript. As suggested, we have developed the quality of English language throughout the whole manuscript. We hope that the above changes can improve the level of our manuscript to meet the standards of journal publication.

Reviewer 2 Report

Comments and Suggestions for Authors

Dear Authors,

The present study, titled "Study on the Function of SlWRKY80 in Tomato Defense against Meloidogyne incognita," investigated the role of SlWRKY80, a tomato gene, in plant defense against the root-knot nematode Meloidogyne incognita (M. incognita). The research demonstrated that overexpression of SlWRKY80 significantly enhanced tomato resistance to this parasitic nematode. This enhanced resistance was accompanied by positive changes in plant growth parameters and potentially involved alterations in plant defense mechanisms mediated by signaling molecules and reactive oxygen species. These findings highlight SlWRKY80 as a promising target for breeding programs aimed at developing tomato varieties with improved resistance against root-knot nematodes.

This study highlights the significance of SlWRKY80 as a positive regulator in tomato defense against the root-knot nematode (M. incognita). The research provides compelling evidence through: (1) Strategic selection of SlWRKY80 based on differential expression in disease-resistant materials; (2) Functional validation using RNAi and overexpression techniques; (3) Physiological evidence linking SlWRKY80 overexpression to enhanced plant growth, soluble protein and sugar content, and root activity; (4) Signaling pathway insights suggesting SlWRKY80's role in modulating SA, JA, and ET signaling; (5) Reduced nematode invasion potentially mediated by ROS accumulation and (6) These findings strongly support SlWRKY80 as a promising target for breeding tomato varieties with improved resistance to root-knot nematodes.

This manuscript offers a well-structured and insightful exploration of SlWRKY80's role as a positive regulator in tomato defense against root-knot nematodes (Meloidogyne incognita). The research employs robust methodologies and provides compelling evidence supporting the significance of SlWRKY80. Based on the study's strengths and its potential contribution to the field, I strongly recommend publication without revisions.

Congratulations to the authors for their commendable work on this subject.

Author Response

Dear Editor and Reviewer2:

Thank you for your letter and for the reviewer’s comments concerning our manuscript entitled “Study on the function of SlWRKY80 in tomato defense against Meloidogyne incognita”. (ID: ijms-3111864).

Thanks again to the reviewer’s recognition of this research and manuscript writing.

Round 2

Reviewer 1 Report

Comments and Suggestions for Authors

The authors addressed most of my concerns; however, a few points still fell short of my expectations.

1. The authors claimed that approximately 50% silencing of SlWRKY80 is effective, citing an example from a study by Tao Wang et al. (2024) where SlNAC3 was shown to suppress cold tolerance in tomatoes by enhancing ethylene biosynthesis (Plant, Cell & Environment, 47(8), 3132-3146). However, there are a few concerns. Firstly, different proteins may require different levels of suppression to achieve their functional outcomes. How did the authors ensure that SlWRKY80 and SlNAC3 require the same level of silencing? Supporting data or references should be provided to clarify this. Secondly, why did the authors not use a knockout or mutated strategies to create a SlWRKY80-deficient strain? If such an approach was attempted, the authors should compare the results of silencing and knockout experiments to demonstrate that both methods yield similar levels of effectiveness.

2. Line 56: Please provide the full name for the 'PR' gene. Additionally, ensure that the full names of all abbreviations are provided when they are first introduced.

3. I understand why the authors discussed the levels of various growth indices in this manuscript. However, the different result sections still lacked a logical connection, making the manuscript appear fragmented.

Comments on the Quality of English Language

need to improve

Author Response

Dear Editor and Reviewer1:

Thank you for your letter and for the reviewer’s comments concerning our manuscript entitled “Study on the function of SlWRKY80 in tomato defense against Meloidogyne incognita”. (ID: ijms-3111864). Those comments are all valuable and very helpful for revising and improving our paper, as well as the important guiding significance to our researches. We have studied comments carefully and have made correction which we hope meet with approval. Revised portion are marked in yellow in the paper. The main corrections in the paper and the responds to the reviewer's comments are as flowing:

1.The authors claimed that approximately 50% silencing of SlWRKY80 is effective, citing an example from a study by Tao Wang et al. (2024) where SlNAC3 was shown to suppress cold tolerance in tomatoes by enhancing ethylene biosynthesis (Plant, Cell & Environment, 47(8), 3132-3146). However, there are a few concerns. Firstly, different proteins may require different levels of suppression to achieve their functional outcomes. How did the authors ensure that SlWRKY80 and SlNAC3 require the same level of silencing? Supporting data or references should be provided to clarify this. Secondly, why did the authors not use a knockout or mutated strategies to create a SlWRKY80-deficient strain? If such an approach was attempted, the authors should compare the results of silencing and knockout experiments to demonstrate that both methods yield similar levels of effectiveness.

Response: Thanks very much for the reviewer's questions. First of all, the 50% silence efficiency we propose here is based on previous studies and the operational experience of our research group over the years. That is, at least 50% silence efficiency can be achieved before subsequent functional verification. In Figure1, it can be clearly seen that the silencing efficiency of most plant individuals can reach 70%-80%. Based on this standard, we selected 10 plants with the lowest silencing efficiency for subsequent tests (mentioned in the method section, line 398), which met the standard. Secondly, the characteristics of various genes are different, and when we do the silencing test of other genes, some silencing efficiency can reach more than 90%, so, as the reviewer mentioned, Crispr-Cas9 verification strategy is the best. However, the method strategy of this study is to rely on the preliminary verification of VIGS combined with overexpression means for deep functional verification, so we did not perform gene editing.

2.Line 56: Please provide the full name for the 'PR' gene. Additionally, ensure that the full names of all abbreviations are provided when they are first introduced.

Response: Thanks very much for the reviewer's query. We added the full name of the ‘NPR gene’ as requested (line 56). We have also added the full names of some words when they are first mentioned (line 79-80, line 219-220).

3.I understand why the authors discussed the levels of various growth indices in this manuscript. However, the different result sections still lacked a logical connection, making the manuscript appear fragmented.

Response: Thanks to the reviewer for the valuable suggestions. We added link descriptions to these parts to make the logical connection more reasonable (line 169-171, line 180, line 199 and line 206-209).

Thanks again to the reviewer for taking time out of their busy schedules to review our manuscript. As suggested, we have developed the quality of English language by AJE throughout the whole manuscript. We hope that the above changes can improve the level of our manuscript to meet the standards of journal publication.